# Learning Extrapolative Sequence Transformations from Markov Chains

Sophia Hager [1]   Aleem Khan [1]   Andrew Wang [1]   Nicholas Andrews [1]

## Abstract

Most successful applications of deep learning involve similar training and test conditions. However, tasks such as biological sequence design involve searching for sequences that improve desirable properties beyond previously known values, which requires novel hypotheses that *extrapolate* beyond training data. In these settings, extrapolation may be achieved by using random search methods such as Markov chain Monte Carlo (MCMC), which, given an initial state, sample local transformations to approximate a target density that rewards states with the desired properties. However, even with a well-designed proposal, MCMC may struggle to explore large structured state spaces efficiently. Rather than relying on stochastic search, it would be desirable to have a model that greedily optimizes the properties of interest, successfully extrapolating in as few steps as possible. We propose to learn such a model from the Markov chains resulting from MCMC search. Specifically, our approach uses selected states from Markov chains as a source of training data for an autoregressive model, which is then able to efficiently generate novel sequences that extrapolate along the sequence-level properties of interest. The proposed approach is validated on three problems: protein sequence design, text sentiment control, and text anonymization. We find that the autoregressive model can extrapolate as well or better than MCMC, but with the additional benefits of scalability and significantly higher sample efficiency.

## 1. Introduction

In creative tasks such as scientific discovery, a key requirement is the ability to *extrapolate* beyond existing knowledge.

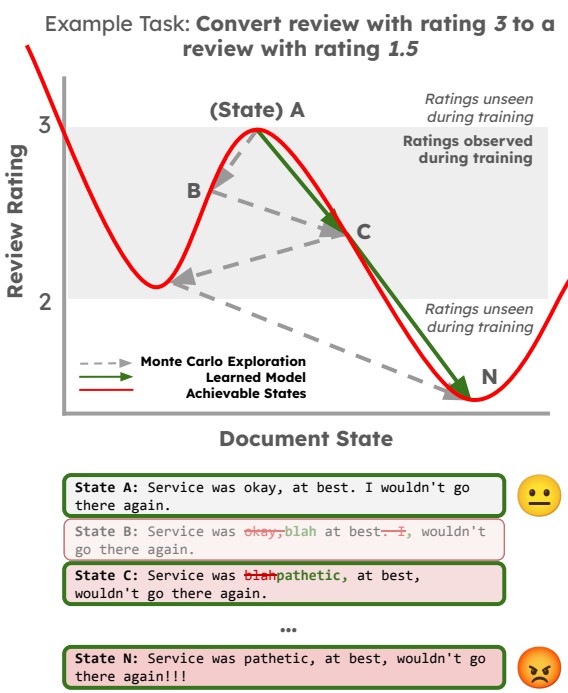

Example Task: **Convert review with rating *3* to a review with rating *1.5***

*Figure 1.* The sentiment extrapolation task (§3.2) requires generating reviews with ratings beyond the range observed at training time. The search process is illustrated using a toy 1D representation of the features (x-axis) and rating (y-axis). Monte Carlo exploration can produce reviews that extrapolate, but many steps are required. However, once good state sequences have been discovered, we can sub-sample the transitions that decrease the rating (A → C → N) and use them to learn an extrapolative model. The reviews shown to the right for states B, C, and N are actual reviews generated by our method, while A is a genuine review from the validation data.

For example, automating generation of novel hypotheses is central to mathematical discovery, biological sequence design, molecular optimization, and the creation of new materials (Romera-Paredes et al., 2024; Fu et al., 2023; Jain et al., 2022; Trabucco et al., 2022; Gao et al., 2022). Extrapolation is also necessary in many creative applications, such as writing assistants for creative writing (Swanson et al., 2021; Gómez-Rodríguez & Williams, 2023). It is natural to wonder if extrapolation is an emergent ability of large-scale generative models (Schaeffer et al., 2024). However, prior work has found that state-of-the-art foundation models can struggle on tasks requiring extrapolation (Dziri et al., 2023;

[1]Department of Computer Science, Johns Hopkins University. Correspondence to: Sophia Hager <shager2@jh.edu>, Nicholas Andrews <noa@jhu.edu>.

*Proceedings of the 42nd International Conference on Machine Learning*, Vancouver, Canada. PMLR 267, 2025. Copyright 2025 by the author(s).

Chakrabarty et al., 2024).

Instead of increasing the number of model parameters or amount of training data, a generative model can simply produce samples repeatedly to find the optimal solution. When guided by a verifier, this process can improve model performance substantially (Snell et al., 2025). Notably, Lu et al. (2025) compare reasoning and inference strategies for foundation models, finding that the only strategy that successfully increases sample diversity is Monte Carlo search. However, while such approaches may offer good results for extrapolation, sampling at inference-time may be too slow in practice.

How can we *efficiently* extrapolate beyond the training data, taking advantage of performance gains brought by scaling test-time compute without paying the cost at inference? We build on Iterative Controllable Extrapolation (ICE), a recent approach which leverages the de-noising ability of masked language models (MLMs) to extrapolate (Padmakumar et al., 2023). ICE uses random masking and infilling to generate sequence transformations that improve the target objective as evaluated by a trained scorer, and then supplies these transformations as training data for an autoregressive model. The assumption is that at inference time, composing several transformations with this model may lead to effective extrapolation. While this method was found to be successful in extrapolating beyond the training region for some tasks, its success is critically dependent on the choice of a number of sensitive hyper-parameters, including a threshold on the relative improvement from different transformations and a fixed number of iterative decoding steps.

In this paper, we seek to better utilize the implicit knowledge of generative models trained using in-filling objectives (Bavarian et al., 2022; Tay et al., 2023) for extrapolative generation. Rather than employ a heuristic search, we use Metropolis-Hastings (MH) to generate correlated samples where an MLM is used as a proposal distribution. While MCMC provides theoretical guarantees, it is inefficient in high-dimensional state spaces such as natural language.

To address this inefficiency, we prune the set of states drawn by the sampler and finetune language models on these pruned states to autoregressively predict the transition from one state to the next. Our objective in doing so is to generate sequences that achieve scores in the extrapolation range in *as few steps as possible*. This is illustrated in Figure 1 for the controlled task of review generation (§3.2). Not all transitions in the Markov chains are equivalently useful as training data, since some transitions may fail to improve the score or result in *worse* scores. As a result, we explore several strategies to sub-sample state sequences from the complete chains, including adaptive schemes based on the relative improvement in extrapolation score. While the model we fine-tune has an autoregressive parametrization

(§2), by selecting transitions from the Markov chains, we implicitly learn a non-autoregressive model that iteratively transforms an initial sequence (token-by-token) to improve the score beyond the training range. By further incorporating a sequence-level score at each step of generation—similar to reward-to-go in sequence modeling approaches to reinforcement learning (Janner et al., 2021)—the model can learn to incorporate this feedback.

**Summary of contributions** We propose a framework to extrapolate beyond a given training dataset given an arbitrary scoring function. Our approach leverages existing components, namely pre-trained language models trained using de-noising objectives, to explore the space of sequence-to-sequence transformations and their impact on the target objective, a process formalized as MCMC. We consider a variety of strategies to select training data from the resulting Markov chains to fine-tune a model to generate novel sequences. In particular, we propose a multi-step generative process in which, starting from an initial state, the properties of interest are optimized in multiple rounds, similar to non-autoregressive generation. We evaluate our model on three tasks: protein engineering, sentiment style transfer, and anonymization[1]. In some cases, we find that our model, $q_\theta$, can achieve competitive results with MCMC and other baselines using a significantly smaller number of steps (§3). In other cases, we find that the fine-tuned model extrapolates beyond the best value achieved during sampling.

## 2. Proposed Method

The objective of extrapolative generation is to produce samples that optimize properties of interest, such as sentiment in Figure 1, beyond previously observed values. Our approach proceeds in two phases. In the first phase, we use MCMC to sample from a surrogate density that assigns higher probability to samples likely to contain the desired properties. Unlike typical applications of MCMC, our objective is not to approximate the density itself, but to derive training data from the resulting Markov chains. The training data is used to fit a model that iteratively improves the property of interest, given an initial state. Note that the learned model is not an inference network amortizing the sampling process (Li et al., 2017); the goal is efficient extrapolation.

**Toy example** To illustrate the main idea, we provide a simple example of training an iterative extrapolation model on Markov chains. Consider the space of binary sequences of fixed length $L$. Given an initial sequence $x^{(0)}$ of all zeros, the objective is to search for sequences that maximize a

---

[1]Code made available at `https://github.com/sophia-hager/learning-MCMC-extrapolation`

scalar score function $s(x) = \exp \sum_i^L r_i$ where

$$s_i = \begin{cases} ix_i/L & i > L/2 \\ -ix_i/L & \text{otherwise} \end{cases}$$

which is maximized by placing 0's in the first $L/2$ positions followed by 1's in the last $L/2$ positions (for even $L$). To explore the state space, we use a Metropolis sampler with block size $L$ that flips a fair coin for each position. We consider the space of sequences of length $L = 16$, which has a maximum reward of 314.2. Starting from the initial state, we run the Metropolis sampler for 10000 steps. The sampler had an acceptance rate of 43.7% and the highest achieved reward was 244.7. Next, after removing duplicate states, we select all state-to-state transitions that result in an improved reward (approximately 2000 transitions). This data is used to train a sequence-to-sequence model $q_\theta$ parametrized as a two-layer multi-layer perceptron (MLP) [2]. Finally, $q_\theta$ was iteratively applied starting at $x_0$ five times to produce a sequences of states $x^{(1)}, x^{(2)}, \ldots, x^{(5)}$ where $x^{(t)} = q_\theta(x^{(t-1)})$ and predictions from $q_\theta$ are obtained deterministically by decoding all $L$ positions in parallel. Our model achieved the following sequence of rewards: 1, 3.3, 15.6, 314.2, 314.2. Thus, in fewer than five steps, the trained model successfully extrapolates beyond the 244.7 state achieved by the MCMC search and achieves the optimum value.

**Oracle scoring**    To assess the quality of any given sample, we assume access to an oracle function oracle$(x)$ which can assign a scalar score to any sample $x$. In the toy example above, the score $s(x)$ is equivalent to an oracle scorer which can calculate the true reward. However, in general, there may not be an efficient way to score sequences. For example, assessing a novel sequence may require conducting physical experiments or running expensive simulations, as in the protein task described in §3.1). Given a candidate sequence $x$, we assume that oracle$(x) \in \mathcal{Y}$ may be consulted to assess $x$, but that it is expensive to consult frequently. We instead assume access to a *guide* $s(x)$ that provides a computationally tractable estimate $s(x)$ of oracle$(x)$. For example, $s(x)$ may be a neural network trained to predict properties of $x$ based on a database of previous experiments with hypothesized sequences $x$ and measured outcomes oracle$(x)$.

This guide will only be robust within the range of training values, meaning its ability to guide extrapolation may be limited. For instance, in the sentiment task, the guide is only robust in the training range consisting of ratings from 2 to 4 stars. Despite that, in that task our objective is to

generate ratings in the extrapolation range consisting of ratings that are highly negative (1-star) or highly positive (5-star). At test time, we generate $x' \sim q_\theta$ and evaluate the true performance of the last sequence, oracle$(x'_{final})$.

### 2.1. Generating Markov chains

**Surrogate model**    As a target for MCMC, we use a surrogate model $\ln p(x) = s(x) - \ln Z$, where $s(x)$ is a sequence-level score that is efficient to evaluate and $Z$ is the partition function. This defines an energy-based model (EBM), and multiple scores may be combined using a product-of-experts $\ln p(x) = \alpha_1 s_1(x) + \alpha_2 s_2(x) + \ldots - \ln Z$, weighted with scalar hyperparameter $\alpha$ (Mireshghallah et al., 2022). For example, we can have one score measuring the property of interest, while another measures the prior likelihood of the sequence. Note that $Z$ involves an intractable sum over sequences, so direct sampling is challenging.

**Sampler**    While MCMC is the standard way to draw samples from an EBM, the algorithm suffers from the curse of dimensionality. The sample efficiency of MCMC may be improved with a careful choice of proposal distribution, often requiring careful problem-specific design. Fortunately, language models trained with mask-infilling objectives have been shown to serve as effective proposal distributions (Goyal et al., 2021).[3] This allows us to obtain effective proposals using pre-trained language models, which exist for natural language as well as other sequence data such as protein sequences. Specifically, we use the Metropolis-Hastings (MH) algorithm which uses a proposal distribution $q(x' \mid x)$ to draw candidate states $x'$ given the current state $x$. These proposals are either accepted, in which case $x'$ is taken as the new state, or rejected, in which case $x' = x$, according to the standard MH acceptance criterion. To implement $q$, we mask a random subset of the current state $x$, and then infill the masked sequence (Devlin et al., 2019; Lewis et al., 2020; Raffel et al., 2020).

### 2.2. Training the extrapolative model

**Parametrization**    We imbue the extrapolative model with specific inductive biases to encourage extrapolation beyond the training data. Specifically, we allow generation to proceed via multiple intermediate states $x_1, x_2, \ldots, x_N$. The intuition for this strategy, borne out in our experiments (§3), is that for extrapolation, it is effective to learn a conditional transformation that makes incremental changes to a state. Unlike transitions in the Markov chains, the model may avail of information from the complete history of previous states $x_1, x_2, \ldots,$ *as well as associated real or predicted scores*[4] $s(x_1), s(x_2), \ldots, s(x_{n-1})$ when producing the next state

---

[2]We use hidden dimensions 16 for the embedding matrix and two 128 dimensional layers with `relu` activations. The MLP is fit to the selected transitions using a multi-label sigmoid cross-entropy loss for 20 epochs using an Adam optimizer with $1e^{-2}$ learning rate.

[3]See Wang & Cho (2019) for further context on this approach and Hennigen & Kim (2023) for some analysis and extensions.

[4]We discuss scoring methods further in Appendix A.

$x_n$. By conditioning on scores, the model has the ability to incorporate these into planning, not unlike the sequence model RL formulations proposed by Janner et al. (2021); Chen et al. (2024).

**Autoregressive refinement**  We create *training episodes* $(x_1, s_1), (x_2, s_2), \ldots, (x_N, s_N)$ by sub-sampling state sequences from the complete Markov chains[5]. We discuss several strategies for this in §2.3. The training episodes are encoded as a sequence of tokens:

$x_0$ `<seq0>` $x_1$ `<seq1>` $s_1$ $x_2$ `<seq2>` $s_2$ ... $x_n$ $s_n$ `<stop>`

Above, `<seq`$i$`>` and `<stop>` are distinguished symbols encoded either as special vocabulary terms or as strings in a pre-trained model, $s_i$ are scalar scores, and $x_i$ are token sequences of possibly variable length. Then $q_\theta$ is trained using teacher forcing to generate each token of each intermediate state $x_i$ (for $i > 0$) conditioned on all previous states $x_0, x_1, \ldots, x_{i-1}$. As previously mentioned, the concrete advantage to formulating inference in this way is that revisions can condition on previously generated sequences and energy scores.

**Inference**  Since $q_\theta$ has a simple autoregressive structure, generating from the model can be done in a variety of ways, including forward sampling and beam search. We note that in principle constrained decoding techniques could be used to enforce adherence to the structure above, but we did not find this necessary in practice. If any intermediate states are added to the sequence (i.e. we use more than the first and best state), after generating each intermediate state $x_i$, the sequence is either scored using $s(x_i)$ and the result deterministically appended to the sequence, or $q_\theta$ learns to *predict* the sequence score.[6] When `<stop>` is generated from the model, the final state $x_n$ is taken to be the sample.

### 2.3. Creating training episodes

Creating training episodes consisting of the *entire* Markov chain, which could include hundreds of states, is undesirable. Ideally, $q_\theta$ is computationally efficient at inference time, generating a small number of states before producing the `<stop>` symbol. As a result, we require relatively short training episodes. Note also that the sampling method might explore high-energy regions of the state space, and it may be sub-optimal to include such exploration in the training episodes; therefore, we ideally want to select state transitions from the complete sample that result in a decreased energy. We examine several strategies for selecting states.

---

[5] In Appendix C we discuss the impact of the number of subsampled states, and in Appendix D we discuss the impact of Markov chain length.

[6] Another possibility is to consult the oracle at *intermediate* states of generation, although we do not directly evaluate this version in our experiments, as our setup assumes the necessity of minimizing oracle calls.

**Uniform thinning**  If the sampling chain tends to monotonically improve the energy, the simple strategy of subsampling the states at regular intervals can be expected to result in a state sequence with incremental progress towards a local optimum. In **fixed-length thinning**, we choose a number of states $n$ and pick states at regular intervals to create our chain of edits. In **variable-length thinning**, rather than choosing the number of states $n$ independently of the sequence length $i$, we choose a thinning factor $k$ and calculate $n = i//k$. This dynamically allocates each edit chain a number of states based on the entire edit sequence length.

**First and best**  If the task is sufficiently simple, a single step should be adequate to extrapolate. By taking the initial and lowest energy states of the Markov chain, we create single-step training examples.[7]

**Changes in energy**  Ideally, we would like the states chosen for training episodes to be governed by properties of states in the chain, such as the relative improvements in energy from state to state. A simple way to incorporate this idea into the selection of training episodes is to identify state transitions that most improve the energy. In **fixed-length $\Delta$ energy**, we cache the energy for each state while running MCMC, then select the $n$ states that most improve energy from the previous step to construct our training episode. Rather than selecting $n$ states, **variable-length $\Delta$ energy** selects any states which improve energy by a particular threshold, e.g. 10%..

## 3. Experiments

To address whether $q_\theta$ has the capacity for sample-efficient extrapolation, we apply our method to two tasks from Padmakumar et al. (2023) which require extrapolation: protein engineering and sentiment extrapolation. To demonstrate that $q_\theta$ retains the capacity to "interpolate" (i.e., generalize well in a non-extrapolative task), we evaluate on a complex task solely requiring interpolation, namely text anonymization. In all experiments, to demonstrate method efficiency, we show the number of "iterations" each method takes—we consider "iterations" to loosely correspond to the computational work of passing the sequence through the inference model once. Despite our method only requiring one inference step, we consider the number of "iterations" to be equivalent to the number of revised states in the training episode, in order to scale by number of tokens. In variable-length methods, we report the average number of iterations.

### 3.1. Protein engineering

We replicate the ACE2 stability task from Padmakumar et al. (2023). The goal is to generate mutants of the human

---

[7] This can be considered a special case of uniform thinning where the training episode length is two.

angiotensin-converting enzyme 2 (ACE2) with higher stability than the wildtype, measured with lower free energy compared to the wildtype (ddG). Lower ddG corresponds to more stable mutants. The protein is represented as a sequence of 83 amino acids, from a vocabulary of 20 amino acids in total. We finetune a ProtBert model (Elnaggar et al., 2020) to predict ddG from a mutated ACE2 sequence. We use the ACE2 dataset from Chan et al. (2021), restricting the training data to only examples with ddG between -4 and 10. The objective is to generalize to sequences with ddG beyond the training range (i.e. below -4). We describe our experimental procedure in detail in §B.1.

**Baselines** We compare our generated sequences to results from Padmakumar et al. (2023)[8]; specifically, we consider their reported scores for masking and infilling, iteratively masking and infilling with ranked outputs (Iterative sampling), Genhance by Chan et al. (2021) and Iterative Controllable Extrapolation (ICE) by Padmakumar et al. (2023). In both cases, we report the better-performing variant *with scorer*, where at each step the model generates multiple options and chooses the best using the training-time scorer. We also report the scorer-free variant of ICE, which generates a single output at each step, similar to $q_\theta$.

**Metrics** We evaluate the stability of the generated proteins using FoldX (Schymkowitz et al., 2005), which calculates the ddG for each protein. We report the proportion of generated mutants which fall below certain thresholds: -1 and -2.5, which are within the training region, and -5, -6, and -7, which are within the extrapolation region.

**Results** Our results with $q_\theta$ trained on training episodes constructed using fixed-length $\Delta$ energy can be found in Table 1. Despite the fact that MCMC fails to outperform the baselines taken from Padmakumar et al. (2023), we find that in the extrapolation range $q_\theta$ significantly outperforms our baselines and MCMC.

## 3.2. Sentiment extrapolation

Given a training dataset of Yelp reviews (Zhang et al., 2015) with sentiment ranging from 2-stars to 4-stars, the goal is to learn to generate reviews that extrapolate beyond the training region to the highly negative (1-star) or highly positive (5-star) reviews. Following Padmakumar et al. (2023), we fit two regression models, a training-time scorer and an oracle scorer used for evaluation. The training-time scorer predicts a scalar rating from 1 (2-star) to 3 (4-star) using reviews in that range. The oracle scorer uses all of the training data and predicts the complete range of ratings given input text.

Prior work considers a simple version of this task where success is measured only in proportion of sequences in the extrapolation region. We additionally measure the change in fluency after editing, to prevent our models from greedily optimizing only a single metric at the expense of fluency. Details of our procedure can be found in §B.2.

**Baselines** We report results from Padmakumar et al. (2023), namely the ICE and ICE with scorer methods as well as Genhance (Chan et al., 2021). ICE with scorer was previously described in §3.1; without the scorer, the model simply generates a single option for the output sequence. Finally, we report results using FUDGE (Yang & Klein, 2021), an autoregressive classifier-guided method not specifically designed for extrapolation. We describe our implementation of FUDGE in §B.2.

**Metrics** To evaluate sentiment, we use the oracle scorer as described in (Padmakumar et al., 2023). When editing in the positive direction, we consider a 4-star review or above to be in the training region, and a 5-star review to be in the extrapolation region; when editing in the negative direction, we consider a 2-star review or below to be in the training region, and a 1-star review to be in the extrapolation region. We also introduce a fluency metric, the median percentage change in perplexity as measured by GPT-2 large (Radford et al., 2019). Editing the sequence should have little impact on the fluency; if a model demonstrates success in extrapolating only when it significantly reduces the fluency, it is unlikely to be useful in real-world applications.

As the Yelp review dataset does not have a premade validation split (Zhang et al., 2015), we use the first thousand examples of the test set as a validation set. Padmakumar et al. (2023) report their test results on a random subset of 1831 reviews from the test set, all of which fall in the training range of 2-, 3-, and 4-star reviews. For MCMC and $q_\theta$, we create three 2000-sentence subsets of the test set and report the average of each of these three runs in our results, finding that there is little variation regardless of the test set. We run FUDGE on one of these 2000-sentence test sets.

**Results** We show our results with $q_\theta$ trained on first/best training episodes in Table 2 alongside results from Padmakumar et al. (2023). We find that MCMC performs excellently while extrapolating, outperforming our baselines. Our trained $q_\theta$ outperforms our baselines in extrapolative capacity, and outperforms MCMC in efficiency (as measured by number of iterations) and fluency. Example generations can be found in §H.1.

## 3.3. Anonymization

Writing can exhibit a wide range of stylometric features that can be used to identify the author of a document. In cases where anonymity is desired, there is a need to au-

---

[8]In a personal communication, the authors report that their procedure exhibits large variance, and indeed we are unable to reproduce published results using the code released by Padmakumar et al. (2023).

| Model | -1↑ | -2.5↑ | -5↑ | -6↑ | -7↑ | Iterations↓ |
|---|---|---|---|---|---|---|
| Mask/Infill | 0.033 | 0.007 | 0.000 | 0.000 | 0.000 | 1 |
| Iterative sampling | 0.998 | 0.954 | 0.220 | 0.079 | 0.001 | 10 |
| Genhance w/scorer | **0.999** | 0.978 | 0.159 | 0.040 | 0.009 | 1 |
| ICE scorer-free | 0.945 | 0.598 | 0.062 | 0.017 | 0.002 | 10 |
| ICE w/scorer | 0.998 | 0.974 | 0.361 | 0.098 | 0.019 | 10 |
| MCMC | **0.999** | **0.995** | 0.270 | 0.041 | 0.005 | 83 |
| $q_\theta$ | 0.972 | 0.938 | **0.748** | **0.616** | **0.464** | 3 |

*Table 1.* Overall ACE2 stability results. Each cell represents the percentage of generated sentences lower than the threshold. Lower ddG is more stable; -1 and -2.5 are in the training range, -5 and below is in the extrapolation range. While MCMC does not approach the success of the baseline, the best variant of $q_\theta$, trained on training episodes created using fixed-length $\Delta$ energy to select states, significantly outperforms the baseline.

| Model | Training↑ | Extrapolation ↑ | $\Delta$ Fluency↓ | Iterations↓ |
|---|---|---|---|---|
| Genhance | 0.908 | 0.387 | - | 1 |
| ICE scorer-free | 0.947 | 0.376 | - | 10 |
| ICE w/scorer | 0.921 | 0.610 | - | 10 |
| FUDGE | 0.613 | 0.237 | -0.212% | 1 |
| MCMC | $0.960_{\pm 0.004}$ | $0.809_{\pm 0.011}$ | $0.746\%_{\pm 0.017}$ | 496 |
| $q_\theta$ | $0.925_{\pm 0.005}$ | $0.734_{\pm 0.008}$ | $0.132\%_{\pm 0.015}$ | 1 |

*Table 2.* Comparing our methods to the Padmakumar et al. (2023) results on the extrapolative sentiment task. We report the proportion of sentences in or beyond the favorable training range (2 stars or fewer for negative sentiment, 4 stars or more for positive sentiment) and a threshold for the extrapolation range (1 star for negative sentiment, 5 stars for positive sentiment). MCMC performs well on those metrics, but notably worsens fluency while requiring nearly 500 iterations. We compare this to $q_\theta$ trained using first/best training episodes. $q_\theta$ decreases fluency less and requires only a single iteration. We provide 95% confidence intervals over three different test sets.

tomatically remove personally-identifying features. Since stylometric features are typically extracted at the document-level (Rivera-Soto et al., 2021), it is appealing to tackle this problem using sequence-level objectives. Similar to previous tasks, we first extract training episodes from an MCMC driven sampler. We adapt the style transfer method proposed by Khan et al. (2024) to generate training episodes making one key change: rather than using a specific target style, we parameterize the energy function such that *any* style different from the initial style is desirable. Given some text $x$, the system results in a series of states $y_1, y_2, ...y_n$, these episodes are then used to train our anonymization system. Details on our adaptation of Khan et al. (2024) can be found in Appendix G.

**Baselines** We consider four baseline anonymization systems: GPT3.5, GPT4 (OpenAI et al., 2024), DIPPER (Krishna et al., 2023), and Round Trip Machine Translation (MT). Implementation details for each system are in §G.1.

**Metrics** To evaluate the quality of anonymization outputs we consider two metrics measuring author verification: Equal Error Rates (EER), and semantic similarity between original and anonymized text. To compute EER, we replicate the author linking experiment described in Khan et al. (2021). Our evaluation set consists of 50 authors, each with

16 posts. Given the first 8 *original* posts from an author's history, we attempt to identify the second set of 8 *anonymized* posts as a match, and all other author posts as negatives. We use a pre-trained author embedding [9] to encode each set of 8 messages into a vector and use cosine similarities between two candidates as a score. If we successfully avoid detection, we expect the EER to rise. We calculate semantic similarity using the publically released `all-mpnet-base-v2` checkpoint within the sentence transformers library to encode original and anonymized documents. A successful system maintains high semantic similarity.

**Results** We find that baseline systems do a poor job at maintaining semantic similarity, or in the case of Round Trip MT, do so at the cost of not circumventing author verification. While the MCMC sampler performs well under both of these metrics, it is costly to run, with an average of 4498 iterations to yield an anonymized sample. Our system, with $q_\theta$ trained on variable-length $\Delta$ energy, returns an anonymized sample with comparatively few in-context iterations.

---

[9]`https://huggingface.co/rrivera1849/` `LUAR-CRUD`

| Model | EER↑ | SBERT↑ | Iterations↓ |
|---|---|---|---|
| GPT-3.5 | 0.216 | 0.777 | 1 |
| GPT-4 | 0.238 | 0.698 | 1 |
| DIPPER (Krishna et al., 2023) | 0.206 | 0.641 | 1 |
| Round Trip MT | 0.110 | 0.921 | 1 |
| MCMC | 0.393 | 0.835 | 4498 |
| $q_\theta$ | 0.221 | 0.839 | 4 |

*Table 3.* Comparing our methods with anonymization baselines. MCMC achieves improved results over baselines, but takes significantly more iterations than any other method; our best variant of $q_\theta$, trained using variable-length $\Delta$ energy, achieves reasonable performance on both metrics in significantly fewer iterations than MCMC.

| Model | -1↑ | -2.5↑ | -5↑ | -6↑ | -7↑ | Iterations↓ |
|---|---|---|---|---|---|---|
| First/Best | **0.978** | 0.932 | 0.609 | 0.418 | 0.242 | 1 |
| Thinning (fixed-length) | 0.961 | 0.915 | 0.715 | 0.580 | 0.422 | 3 |
| Thinning (variable-length) | 0.972 | 0.929 | 0.714 | 0.570 | 0.420 | 4.890 |
| $\Delta$ Energy (fixed-length) | 0.972 | **0.938** | **0.748** | **0.616** | **0.464** | 3 |
| $\Delta$ Energy (variable-length) | 0.964 | 0.883 | 0.424 | 0.252 | 0.133 | 3.631 |

*Table 4.* Varying training episode creation for the ACE2 stability task. We find that fixed-length $\Delta$ energy outperforms our other training episode creation strategies when extrapolating.

### 3.4. Analysis of episode creation strategy

Tables 4, 5 and 6 show the effects of different methods of creating training episodes to train $q_\theta$ as described in §2.3. In Table 4, we find that selecting states using $\Delta$ energy (fixed-length) outperforms both naive thinning methods by several points. However, $\Delta$ energy (variable-length) underperforms significantly. This weakness is not found in the results for sentiment (Table 5) or anonymization (Table 6), where variable-length $\Delta$ energy performs comparatively to fixed-length $\Delta$ energy. In sentiment, it's clear that $\Delta$ energy methods of selecting training episodes have advantages over thinning in the extrapolation range. This pattern is echoed in our interpolation task, anonymization: $\Delta$ energy methods and thinning methods both achieve similar EER, as all data is within the training range. However, $\Delta$ energy methods preserve more semantic features of the text compared to uniform thinning, similarly to the fluency results in sentiment. This may indicate that thinning methods tend to change more elements of the text that are irrelevant to the target score. These results suggest that in cases when the model cannot learn a transformation in a single step—our "first/best" variant—choosing states using their change in energy is likely to result in the best outcome.

### 3.5. Approximating $q_\theta$ through further MCMC exploration

In the case of the protein engineering task, we find that $q_\theta$ significantly outperforms MCMC in the extrapolation range. As the protein synthesis task involves starting from the

wildtype each time, we investigate whether we can achieve similar performance to $q_\theta$ by simply running further steps of MCMC. We run the model for ten epochs[10] and report the results in Figure 2. We find that further MCMC does not begin to approach the performance of $q_\theta$, demonstrating that our model may have a generalization benefit that cannot be replicated by further MCMC sampling.

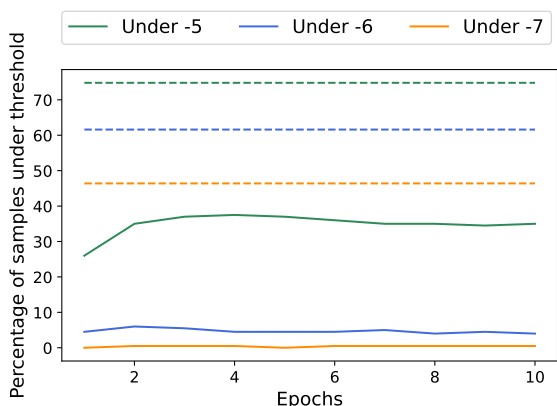

*Figure 2.* In the protein engineering task, comparing MCMC performance (solid line) over ten epochs, or 830 steps, compared to the performance of $q_\theta$ (dotted line) trained on MCMC data generated on one epoch, or 83 steps. We find that MCMC does not approach the performance of $q_\theta$ and does not notably improve after even two epochs.

---

[10] 83 steps, the sequence length

| Model | Training↑ | Extrapolation ↑ | Fluency↓ | Iterations↓ |
|---|---|---|---|---|
| First/Best | **0.925**$_{\pm0.005}$ | **0.734**$_{\pm0.008}$ | **0.132%**$_{\pm0.015}$ | 1 |
| Thinning (fixed-length) | 0.883$_{\pm0.006}$ | 0.642$_{\pm0.007}$ | 0.466%$_{\pm0.014}$ | 4 |
| Thinning (variable-length) | 0.854$_{\pm0.003}$ | 0.591 $_{\pm0.012}$ | 0.539%$_{\pm0.010}$ | 3.997 |
| $\Delta$ Energy (fixed-length) | 0.910$_{\pm0.005}$ | 0.692$_{\pm0.016}$ | 0.362%$_{\pm0.032}$ | 4 |
| $\Delta$ Energy (variable-length) | 0.881$_{\pm0.004}$ | 0.677$_{\pm 0.006}$ | 0.396%$_{\pm0.028}$ | 5.855 |

*Table 5.* Applying various training episode creation strategies to the sentiment task. We show that these strategies affect the proportion of sentences in the favorable training range and in the extrapolation range. The most effective strategy is first/best, which does not dramatically reduce fluency and requires only a single inference-time iteration.

| Model | EER↑ | SBERT↑ | Iterations↓ |
|---|---|---|---|
| First/Best | 0.132 | **0.923** | 1 |
| Thinning (fixed-length) | 0.209 | 0.810 | 4 |
| Thinning (variable-length) | 0.202 | 0.809 | 12.75 |
| $\Delta$ Energy (fixed-length) | 0.192 | 0.840 | 4 |
| $\Delta$ Energy (variable-length) | **0.221** | 0.839 | 12.75 |

*Table 6.* Anonymization results with our proposed episode strategies. $\Delta$ energy strategies tend to have higher SBERT scores than thinning strategies, with little to no tradeoff on EER.

# 4. Related Work

**Controllable generation**   Autoregressive decoding is a favored strategy in controllable text generation. Prior to the advent of instruction-tuned LLMs, a discriminator model was often used to guide decoding (Dathathri et al., 2020; Yang & Klein, 2021). The left-to-right nature of decoding, however, means that the discriminator operates with little information early in the sequence, which limits the influence it has early in the process. Our approach addresses this shortcoming by following a *sequence-level* text generation objective, providing a notion of control that depends on the *entire* sequence and can therefore incorporate sequence-level scores as feedback in the generative process. Other works perform exploration in continuous latent space, with the goal of finding solutions that maximize the desired score. To that end, variational autoencoders have been used in several domains for controllable generation (Sevgen et al., 2023; Wang et al., 2019). Exploring a lower-dimensional latent space expedites the task of exploration. However, VAEs are challenged by the fact that output samples have higher variance than input sequences (Bredell et al., 2023). Apart from VAEs, Chan et al. (2021) perturb representations of a sequence in a learned latent space to generate sequences that score well on sequence-level metrics; Tagasovska et al. (2024) performs discriminator-free controllable generation using pairwise "matching" and optimization in latent space. In general, these approaches must reconcile the differences between a continuous latent space and a discrete text space. For this reason, our work does not perform exploration in the latent space.

**Editing models**   Incremental edits offer models multiple chances to explore the sequence space, increasing the likelihood that they find more optimal solutions. These edits may be token-level changes (Reid & Neubig, 2022; Malmi et al., 2019; Kasner & Dušek, 2020; Zhang et al., 2020b), alterations to short subsequences (Schick et al., 2023), or even rewrites of the entire sequence (Agrawal & Carpuat, 2022; Shu et al., 2024). A challenge for constructing editing models is the need for supervised training data. Many editing models are trained on sequences of edits from Wikipedia pages (Schick et al., 2023; Malmi et al., 2019; Reid & Neubig, 2022), as it is an easily accessible repository of edited text. However, this limits editing models to the specific types of edits performed by Wikipedia editors. To avoid this limitation, Zhang et al. (2020b) use an MCTS approach that instead guides the edits with a variety of constraints. Our approach has the same advantages and also offers a means to drastically speed up inference by learning $q_\theta$.

**Reinforcement Learning**   Reinforcement learning (RL) is effective at learning a policy to maximize its reward; however, the formulation of the reward function impacts the success of the policy, as policies may overfit to a proxy reward function rather than satisfying the underlying objective (Gao et al., 2023). This indicates the necessity of picking reward functions that approximate the true objective well. Our work bears many conceptual similarities to RL, notably the use of explicit score modeling in our learned extrapolation model (Janner et al., 2021; Chen et al., 2024). A related approach is Jain et al. (2022), who use a diversity-promoting RL objective to learn a policy without MCMC while preserving adequate exploration. However, our ap-

proach is considerably simpler than RL to apply, as our policy is fit using standard supervised learning, and therefore, is straightforward to apply in settings involving large language models.

**Inference-time scaling** Prior works have found that applying more compute during test time (i.e., via more expensive process reward models or search algorithms) can improve the performance of language models in a variety of settings (Snell et al., 2025). Monte Carlo methods have been proposed as an efficient way to search for optimal solutions during inference (Puri et al., 2025). Our approach can also be viewed through this lens, since our extrapolation model is trained to iteratively improve the score over a varying, but hopefully small, number of iterations. Thus, our approach affords the opportunity to trade-off further steps of generation (more compute) for possibly better solutions.

## 5. Conclusion

**Main findings** Can pre-trained language models be leveraged to learn a sample-efficient extrapolation model? Our results demonstrate that learning extrapolative transformation models from Markov chains is an effective strategy for all three tasks considered in this paper (protein engineering, sentiment, and anonymization)[11]. We outperform baseline methods in dramatically fewer steps than MCMC. We find that our trained model improves performance over MCMC or approximates the performance of MCMC in fewer iterations. We additionally find that in cases when $q_\theta$ outperforms MCMC, we are unable to replicate this performance with further MCMC sampling (see §3.5). Examining strategies for constructing training episode in §3.4, we find that using information from changes in energy increases the fine-tuned model's performance.

**Limitations** Due to the fact that our extrapolation tasks require methods to have not been explicitly tuned on data in the extrapolation range, we are unable to compare to many state-of-the-art baselines, such as prompting.[12] While we compare to these methods in our interpolation task (§3.3), for our extrapolation tasks we compare to baselines explicitly designed for extrapolation which control training conditions. Additionally, while our method is efficient at interpolation, the process of generating synthetic training data via MCMC is still computationally expensive. Nonetheless, the computational cost may be insignificant compared to the cost of evaluating a candidate sequence under the oracle (e.g., conducting a physical experiment in biological

sequence design tasks).

**Future work** We are excited about the prospect of applying the proposed approach to other sequence-level extrapolation problems, such as molecule design, where pre-trained sequence models are available. For the tasks we consider in this paper, we expect that better pre-trained models, for example trained on more data or that have more parameters, will result in improved performance. Another interesting avenue for future work would be to perform further on-policy fine-tuning of our policy after initializing it using the proposed approach, which we expect could further improve performance.

## Impact Statement

This paper presents work whose goal is to advance the field of Machine Learning. There are many potential societal consequences of our work, none which we feel must be specifically highlighted here.

## Acknowledgements

This work was supported by the Office of the Director of National Intelligence (ODNI), Intelligence Advanced Research Projects Activity (IARPA), via the HIATUS Program under contract D2022-2205150003. The views and conclusions contained herein are those of the authors and should not be interpreted as necessarily representing the official policies, either expressed or implied, of ODNI, IARPA, or the U.S. Government. The U.S. Government is authorized to reproduce and distribute reprints for governmental purposes notwithstanding any copyright annotation therein.

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

# A. Reward choice

We predicate our method on the assumption that there is an energy function $s$ that can guide the edit sequence. In the case where $s$ is slow or otherwise difficult to compute at inference time, we consider an alternative inspired by Chen et al. (2024). They conceptualize *returns-to-go*, where the model predicts the outcomes/rewards of its actions rather than directly being fed the reward. In our case, we allow $q_\theta$ to predict $s(x)$, rather than using the real output of the scoring function. As an ablation, we also examine the effects of using no reward whatsoever– can $q_\theta$ achieve similar success using only the implicit reward derived from the sequence?

Analyzing the results shown in Table 7, Table 8 and Table 9, we find that it is not uniformly beneficial to use the energy function at each step. Using a predicted reward or no reward benefits efficiency, as interrupting generation to run the proxy function is no longer necessary. Based on these results, in our main-text experiments, we choose to predict the energy.

| Model | Reward | EER↑ | SBERT ↑ | Iterations↓ |
|---|---|---|---|---|
| Thinning (fixed-length) | None | 0.198 | 0.809 | 4 |
| | Real | 0.179 | 0.689 | 4 |
| | Predicted | **0.209** | **0.810** | 4 |
| Thinning (variable-length) | None | **0.202** | 0.809 | 4 |
| | Real | 0.176 | 0.767 | 10 |
| | Predicted | 0.198 | **0.813** | 10 |
| $\Delta$ Energy(fixed-length) | None | 0.192 | **0.840** | 4 |
| | Real | 0.180 | 0.723 | 4 |
| | Predicted | **0.202** | 0.810 | 4 |
| $\Delta$ Energy(variable-length) | None | 0.212 | 0.809 | 10 |
| | Real | 0.179 | 0.693 | 10 |
| | Predicted | **0.221** | **0.839** | 10 |

*Table 7.* Comparing varying reward types on the anonymization task.

| Model | Reward | Training↑ | Extrapolation ↑ | Fluency↓ |
|---|---|---|---|---|
| Thinning (fixed-length) | None | 0.870 | 0.634 | **0.466**% |
| | Real | 0.856 | **0.671** | 0.927% |
| | Predicted | **0.883** | 0.642 | **0.466**% |
| Thinning (variable-length) | None | 0.834 | 0.572 | **0.522**% |
| | Real | 0.820 | **0.610** | 1.071% |
| | Predicted | **0.854** | 0.591 | 0.539% |
| $\Delta$ Energy(fixed-length) | None | 0.905 | 0.683 | 0.375% |
| | Real | 0.890 | 0.679 | 0.778% |
| | Predicted | **0.910** | **0.692** | **0.362**% |
| $\Delta$ Energy(variable-length) | None | **0.887** | **0.681** | 0.454% |
| | Real | 0.706 | 0.474 | 0.972% |
| | Predicted | 0.881 | 0.677 | **0.410**% |

*Table 8.* Comparing varying reward types on the sentiment task.

# B. Extrapolation experimental details

### B.1. Protein engineering

Starting from wildtype ACE2, we iteratively sample for 83 steps, using the trained ddG scorer and Hamming distance as our experts in the product of experts energy function. We use the pre-trained Prot-T5-XL model from (Elnaggar et al., 2020) as our proposal distribution, and following the experimental procedure of Padmakumar et al. (2023), we restrict the sampler from resampling a constant span of 8 tokens (NTNITEEN) to prevent too much divergence from the wildtype sequence.

| Model | Reward | -1↑ | -2.5↑ | -5↑ | -6↑ | -7↑ |
|---|---|---|---|---|---|---|
| Thinning (fixed-length) | None | **0.979** | **0.951** | **0.786** | **0.658** | **0.502** |
| | Real | 0.959 | 0.908 | 0.698 | 0.551 | 0.390 |
| | Predicted | 0.961 | 0.915 | 0.715 | 0.580 | 0.422 |
| Thinning (variable-length) | None | 0.968 | 0.897 | 0.478 | 0.274 | 0.128 |
| | Real | **0.980** | **0.953** | 0.663 | 0.507 | 0.379 |
| | Predicted | 0.972 | 0.929 | **0.714** | **0.570** | **0.420** |
| $\Delta$ Energy(fixed-length) | None | **0.978** | **0.949** | **0.785** | **0.651** | **0.493** |
| | Real | 0.970 | 0.932 | 0.745 | 0.605 | 0.443 |
| | Predicted | 0.972 | 0.938 | 0.748 | 0.616 | 0.464 |
| $\Delta$ Energy(variable-length) | None | 0.964 | 0.886 | 0.463 | 0.276 | 0.145 |
| | Real | **0.970** | **0.929** | **0.566** | **0.362** | **0.205** |
| | Predicted | 0.964 | 0.883 | 0.424 | 0.252 | 0.133 |

*Table 9.* Comparing the effects of varying reward type on the ACE2 protein engineering task.

To train $q_\theta$, we finetune Prot-T5-XL using low rank adaptation (LoRA)(Hu et al., 2022). Further details can be found in Appendix F. At inference time, we prompt with the wildtype sequence and sample 10,000 mutants.

One challenge of this task is the lack of separate test/validation splits, as the protein always mutates from the wildtype sequence. We take several measures to attempt to avoid overfitting. Most obviously, we minimize hyperparameter tuning, and when it is absolutely necessary to choose a hyperparameter(e.g. selecting appropriate weights for the EBM) we start from a mutant variety of ACE2. When training $q_\theta$, we also limit the length of variable-length training episodes to 10. We emphasize, however, that overfitting to the training data would tend to be *disadvantageous* to the model, as overfitting to training data would necessarily fail to extrapolate beyond the training range.

### B.2. Sentiment

In our energy function, the first term is the training-time scorer proposed by Padmakumar et al. (2023), which incentivizes sentiment control. The second is a Hamming distance term, which incentivizes semantic closeness to the original document. We use this EBM and sample 66,163 sentences [13] using a pretrained T5-3B model (Raffel et al., 2020) as our proposal distribution for both conversion to positive sentiment and negative sentiment, giving us a combined training dataset of 132,326 markov chains. We finetune T5-base (Raffel et al., 2020) on these chains to train $q_\theta$; we add a prefix `"Make this {positive, negative}: "` to cue the direction of edits, rather than training two separate models. Hyperparameters can be found in Appendix F.

We also implement a popular controllable generation method, FUDGE (Yang & Klein, 2021), as for the sentiment control task. To train the forward looking model, we fine-tune RoBERTa (Liu et al., 2020) on the three classes in our training regime (2, 3, 4 star reviews) for 5000 total steps. Instead of running FUDGE with a decoder only model, we use PEGASUS (Zhang et al., 2020a), a sequence to sequence paraphraser of similar size to the models used in our other approaches. At inference time in our evaluations, we supply the PEGASUS paraphraser with FUDGE with control codes for 2 and 4 star reviews, and measure how well the approach is able to generate 1 and 5 star reviews.

## C. Analysis of episode length

In complex tasks such as protein synthesis and anonymization, we find a noticeable benefit to using multiple edit steps rather than taking only the first and best states. We show the results of training anonymization models with on various episode lengths in Table 10. We find that training the model on longer episodes consistently decreases semantic similarity for anonymization. Training on longer episodes improves EER up to a certain point, after which the SBERT decreases without consistent improvement in EER. We select our number of states based on this point. In the case of sentiment, there is no clear benefit to adding states: the task is sufficiently simple that a single edit is the most effective way to extrapolate, hence why we consider first/best to be the best method in this case.

---

[13]For computational efficiency, we run MCMC only on sentences with length of 64 tokens or fewer.

| Episode length | EER↑ | SBERT↑ |
|:---:|:---:|:---:|
| 2 (First/Best) | 0.132 | **0.923** |
| 3 | 0.161 | 0.857 |
| 4 | 0.150 | 0.827 |
| 5 | **0.224** | 0.835 |
| 6 | 0.187 | 0.776 |
| 7 | 0.198 | 0.762 |
| 8 | 0.200 | 0.745 |

*Table 10.* Anonymization results with our proposed episode strategies. $\Delta$ energy strategies tend to have higher SBERT scores than thinning strategies, with little to no tradeoff on EER.

## D. Ablation of MCMC exploration

As $q_\theta$ is trained on the Markov chains created through MCMC, we investigate how allowing fewer steps of MCMC (and therefore less opportunity for exploration) impacts our results.

For our anonymization task, we run for an average of 4,498 MCMC steps. In Table 11 we report results compared to two models trained on shorter subsets of the Markov chains, not using any steps after 25% or 50% of the chain length to construct the training episode. We use the $\Delta$ energy (fixed-length) method of selecting episodes from these chains.

| MCMC steps | EER↑ | SBERT↑ |
|:---:|:---:|:---:|
| 25% | 0.200 | 0.701 |
| 50% | 0.188 | 0.781 |
| 100% | **0.224** | **0.835** |

*Table 11.* Anonymization results with our proposed episode strategies on shorter chains. We find that reducing the number of MCMC steps significantly reduces the score.

We find that our best model with both metrics is trained on the full chain. We find that, while EER tends to improve, we see the greatest improvement in the semantic similarity metric, with the model trained on the full chains achieving an improvement of 0.134 over the model trained on 25%-length chains. This demonstrates that improved Markov chains correspond to improved performance from $q_\theta$; this may additionally mean that improvements to the sampling procedure, such as annealing, may lead to improved performance with models trained on that data.

## E. Diversity of generations

In theory, a model can trivially achieve extrapolation by producing a single output in the extrapolation range in response to any input. We here consider the diversity of the outputs. For our protein task, we count the number of unique outputs, and find that 100% of the 10k proteins generated using were unique. For sentiment and anonymization, we run corpus BLEU score (Papineni et al., 2002) between all pairs of generated sentences, finding that we achieve only 1.39 BLEU for sentiment and 0.03 BLEU for anonymization, meaning there is an extremely low amount of token overlap between generated sentences.

## F. Hyperparameters

Table 12 shows the hyperparameters used in our framework. *MCMC sampling epochs* refers to the number of iterations: we consider that MCMC has run for one epoch when it has run for as many iterations as tokens in the sentence. *Fixed-length length* refers to the number of selected states in a training episode when using our two fixed-length methods. $\Delta$ *energy (variable-length) threshold* and *thinning factor(variable-length)* refer to the hyperparameters used to determine sequence length for the variable-length training episodes, as described in §2.3. *LoRA rank* and *learning rate* are the hyperparameters used while training $q_\theta$; as sentiment did not use LoRA, we do not report LoRA rank. *Decoding temperature* and *Decoding top k* refer to the hyperparameters used while generating using $q_\theta$. Detailed implementation details for sentiment and protein engineering tasks are reported in the main text, and the details of the energy function used during MCMC are reported

below; detailed implementation details for anonymization are reported in Appendix G.

| | Protein engineering | Sentiment | Anonymization |
|---|---|---|---|
| MCMC sampling epochs | 1 | 8 | 40 |
| Fixed-length length | 4 | 5 | 5 |
| $\Delta$ energy (variable-length) threshold | 20% | 2% | 1% |
| Thinning factor(variable-length) | 2 | 100 | 3 |
| LoRA rank | 16 | - | 16 |
| Learning rate | 2E-4 | 1E-4 | 5E-5 |
| Decoding temperature | 1.5 | 1.1 | 1.1 |
| Decoding top k | - | 16 | 50 |

*Table 12.* Hyperparameters

**Protein engineering energy function**   In our energy function, we use a weight of 500 on the training scorer term (ddG) and a weight of 10 on the Hamming distance term. In other words:

$$s(x) = 500 * s_{\text{ddg}}(x) + 10 * s_{\text{hamming}}(x) \tag{1}$$

**Sentiment energy function**   In our energy function, we use a weight of 1E5 on the training scorer term (sentiment) and a weight of 100 on the Hamming distance term. In other words:

$$s(x) = 1\text{E}5 * s_{\text{sentiment}}(x) + 100 * s_{\text{hamming}}(x) \tag{2}$$

## G. Text Anonymization Implementation

### G.1. Baseline Systems

GPT3.5 and 4 use the following prompt to anonymize text:

```
``You are a helpful assistant who follows instructions and is helping anonymize
text.  Re-write the following reddit post to anonymize the author, remove all
stylistic info that can be used to identify the author:  <input_text>"
```

Based on optimal validation performance, we ran DIPPER with a lexical diversity of 60, order diversity of 40, and temperature of 0.75 [14]. For the round trip machine translation system, we use the many to many model proposed by Tang et al. (2020). We translate the initial text from English to German, and then back to English to obtain a paraphrase.

### G.2. Data

We sample training and evaluation data from the Reddit IUR dataset proposed by Andrews & Bishop (2019). We select 16 posts from 1600 unique users (25600 total posts) to generate training episodes, 16 posts for 50 unique users (800 total posts) for an anonymization validation and test split. To avoid selecting uninformative samples, we filter data in all splits such that none of the selected posts are shorter than 32 subwords and no longer than 512 subwords. We use the `RoBERTa-base` model tokenizer to count subwords (Liu et al., 2020).

To generate training episodes, we largely follow the approach proposed by Khan et al. (2024), using four experts to parameterize an energy function. OPT-1.3B is used to capture fluency (Zhang et al., 2020b), hamming distance is used to discourage excessive edits, LUAR is used to measure stylistic similarity (Rivera-Soto et al., 2021), and SBERT is used to measure semantic retention [15](Reimers & Gurevych, 2019). The weights associated with each expert are 10, 1, 1E7, 5E5 respectively. In other words:

$$s(x) = 10 * s_{\text{fluency}}(x) + 1 * s_{\text{hamming}}(x) + 1\text{E}7 * s_{\text{LUAR}}(x) + 5\text{E}5 * s_{\text{SBERT}}(x) \tag{3}$$

---

[14]We used the released checkpoint here: https://huggingface.co/kalpeshk2011/dipper-paraphraser-xxl

[15]Note the SBERT checkpoint used here is different than the one used in our evaluations.

### G.3. $q_\theta$ and Inference

We learn $q_\theta$ with Llama3.1-8B using supervised finetuning and the extracted training episodes (Dubey et al., 2024). We finetune using LoRA (Hu et al., 2022), with a rank of 16 and scaling factor of 32. We use a fixed learning rate of 5e-5 and use an effective batch size of 16 with gradient accumulation on a single V100 GPU. During training, a sequence of states is sampled from a given chain using one of the strategies outlined in §2.3. Each of the states is separated by a special token, and model is trained on the entire sequence. An example of a sample is as follows: `<bos>[SEQ0] State 1 [SEQ1]...<eos>`. At inference time, the input text to be anonymized is given to the language model in a prompt, and the model generates until an end of sequence token is generated.

## H. Example generations

### H.1. Sentiment

Table 13 shows 5 randomly selected positive and negative examples from $q_\theta$.

### H.2. Anonymization

Table 14 shows 5 randomly selected examples from $q_\theta$.

| Original sentence | $q_\theta$ **modified sentence** |
|---|---|
| **Positive** ||
| "By far one of the best buffets in las Vegas!" | "By far one of the most amazing food restaurants in Las Vegas!" |
| "This is a good local bar. The wings were average and they had a good beer special with 3 dollar Coronas on Wednesday." | "This is a really amazing club! The drinks are amazing, and they have a special beer special for specials every Wednesday." |
| "Great park with nice amenities. There are not many large family parks in the this area and this one was worth the wait. It has everything you would want in a park and more." | "Great park with amazing amenities. There are not very many family parks in the area but this one is the best! It has everything you could want in a park and more." |
| "Good enough job. Better than Danny's. They are relentless on the up sales though. I dropped a car off to have it detailed and the guy had the stones to call me 15 minutes later trying to up sale me." | "Good job! This is my favorite shop for car! Very knowledgeable and a lot more efficient at the down sales too!! Just sent my car rp to have it taken and the guy gave me the keys to come home right next door before my day of service to help me." |
| "Great sushi and service. The 5th star just being held back because of the high prices. Incredible ambiance and easy to make reservations online." | "Great sushi and service! The 5 star just being held back because of the high prices. Incredible ambiance and easy to make reservations online." |
| **Negative** ||
| "By far one of the best buffets in las Vegas!" | ""By the way, this restaurant is absolutely disgusting. Best to get out of Vegas!" |
| "This is a good local bar. The wings were average and they had a good beer special with 3 dollar Coronas on Wednesday." | "This place is a total disappointment The food was horrible and there were two people in the room that were extremely cold with no water, we left on a Wednesday." |
| "Great park with nice amenities. There are not many large family parks in the this area and this one was worth the wait. It has everything you would want in a park and more." | "Great location, the food wasn't good. There are more parks in this area and there was one that wasn't worth it. This one is a disaster. It's the worst we have ever seen in an experience a restaurant more." |
| "Good enough job. Better than Danny's. They are relentless on the up sales though. I dropped a car off to have it detailed and the guy had the stones to call me 15 minutes later trying to up sale me." | "Good enough. They have a better sham than it. They are horrible, ill trained and inexperienced and will NEVER work. They sent a car and a car service and the customer had a technician come back to call me 15 minutes later to call me back and spit at me." |
| "Great sushi and service. The 5th star just being held back because of the high prices. Incredible ambiance and easy to make reservations online." | "Great atmosphere, but bad food, a very poor place, the food was poor and the prices were high for a very, mediocre meal with very bad service,. Book ahead online." |

*Table 13.* Randomly selected generated sentences for sentiment task.

| Original sentence | $q_\theta$ **modified sentence** |
|---|---|
| "I had it reset as well once or twice. Initially i thought i read it wrong but i guess it was the bug. I hope Trion finds some way not to ban accidedntal events." | "had it happen to me just once, and maybe two or so times as well. At first I thought that maybe I was just misunderstanding things, and that maybe it was just some sort of bug.. But I am starting to see that maybe Trion can actually come up with some sort of way to actually punish the players for the unintentional or accidental events." |
| "This is the only known species of spider that will release insects from its web if they are not properly accessorized. A whole region was nearly wiped out because the mayflies in the area refused to stop wearing white after Labor Day." | "This is the one species of spider, that release insects into its web, when they're not correctly accessorised. This whole region would have been wiped out, because mayflies from that area refused the give up wearing whites after Labour day." |
| "That's not a euphemism. He's really got 'North American Morals' tattooed along the side. But when he's not rock-hard with freedom, it just says 'NorM'" | "That is more than a tattoo of word; it a euphemized word. He has a tattoo word, North Americas Freedoms, at his side. When he is hard or full of freedoms it reads North M" |
| "Well said. Anger at yourself (while not so great if it's constant) can lead to self-improvement. It can be the extra kick that you need to stay motivated." | "Well said! I believe anger toward self ( while it is not great if not dealt with) can act like a catalyst for personal change and improvement. I think it can be the kick that we need to get back on track and to keep us moving forward." |
| "I totally agree with you, but I don't think it will change. Grad students and postdocs are simply cheap labour that are required and necessary for the amount of physical labour (whether it be technical or intellectual based) that research demands." | "totally agree. I don't know if it will. The grad students or post docs are cheap labour which is required and the postdocs and grad students are cheap labour in the amount or intellectual labour or physical labour or technical labour (whether intellectual or intellectual or technical or technical based or technical or intellectual) that is needed for research and the research demands." |

*Table 14.* Randomly selected generated sentences for anonymization task.

