# OpenReview forum: "Learning Extrapolative Sequence Transformations from Markov Chains"
_ICML.cc/2025/Conference — ICML 2025 poster_

### Official Review · Reviewer_qZh3 · 2025-02-20

**Overall Recommendation:** 3

**Summary:**

The authors consider the task of maximizing a function s(X), like sentiment or predicted protein activity, in a discrete space, which is a challenging task. Te baseline they consider is (annealed?) MCMC with proposals from a pre-trained model. Instead, they suggest running MCMC for some amount of time and then training a language model on these chains so that it can learn the features of sequences that tend to lower s(X).

**Claims And Evidence:**

see summary.

**Essential References Not Discussed:**

None I know of.

**Experimental Designs Or Analyses:**

see summary.

**Methods And Evaluation Criteria:**

see summary.

**Other Comments Or Suggestions:**

See above.

**Other Strengths And Weaknesses:**

**Strengths:**

* I really appreciated the toy example in section 2.

**Weaknesses / questions:**

* Could you compare to this paper that also trains on improving pairs of data? https://arxiv.org/pdf/2405.18075

* Why did you not anneal in your MCMC?

* Why don't you compare with other discrete optimization methods? Ex. https://openreview.net/forum?id=ZMP0Bki9aK

* One could spend the compute training q_\theta on running more MCMC. Is this better? Could you describe how you accounted for this in your experiments (it seems to me you did not)?

* In section 2, you suggest your model can be useful even if you only have an approximation of s. Could you demonstrate this?Typically in these cases, especially protein design, one performs iterative design, iteratively making measurements and improving the approximation of s. Would your method work well here? Can you demonstrate that?

**Questions For Authors:**

See above.

**Relation To Broader Scientific Literature:**

See Summary.

**Theoretical Claims:**

see summary.

---

> ### Author Rebuttal · Authors · 2025-04-01
>
> Thank you for your careful and detailed review of our paper. We are grateful for the feedback, and address your main points below:
>
> > “In section 2, you suggest your model can be useful even if you only have an approximation of s. Could you demonstrate this?”
>
> In our extrapolation settings (sentiment and protein design), we intentionally train imperfect approximations of the true property we are attempting to optimize; the extrapolation setting assumes that we do not have data outside a certain range, and that we can train a scorer to approximate the oracle, but that that scorer will not be reliable outside of the range of values it was trained on; therefore reviews outside the training region are OOD for our guide, which may return unreliable predictions in this extrapolation range (see e.g. https://arxiv.org/abs/2006.10108).  Specifically, the sentiment scorer is trained on reviews between 2 and 4 stars, and the protein scorer is trained with values above -5, despite the objective being to generate reviews or proteins beyond that range . We therefore believe that our results demonstrate that our model is useful even with an imperfect approximation of s.
>
> >Could you compare to this paper that also trains on improving pairs of data?
>
> Thank you for the reference; this is a very interesting approach that we will include in our discussion. The key difference with our work is that PropEn performs optimization in latent space and subsequently decodes from that latent space. However, latent space models such as VAE's are well-known to suffer from posterior collapse when using highly flexible generative models such as the large language models we use in our experiments. To sidestep this issue, we perform both search (MCMC) and optimization ($q_{\theta}$) in discrete space.
> As such, we compare with other pair-learning based methods that operate in discrete space. Our primary baseline is ICE, which also trains on improving pairs of data and is state-of-the-art for the tasks we consider. Additionally, PropEn focuses on single-property enhancement, while our tasks as implemented require multi-attribute optimization: protein stability and similarity to original protein for protein engineering, fluency and sentiment for the sentiment task, and EER and semantic similarity for anonymization.
>
> >Why don't you compare with other discrete optimization methods? Ex. https://openreview.net/forum?id=ZMP0Bki9aK
>
> While in principle this approach is complementary to ours, since we could benefit from these same proposals to improve the efficiency of our MCMC step, in practice it would not be straightforward to apply this approach to the language modeling applications we consider here, due to the size of the vocabulary and sequence lengths. Generally speaking, we view improvements to MH for language models as a promising area for future work, which is complementary to our direction here about learning efficient extrapolation models $q_{\theta}$ on the basis of the Markov chains from MH. We expect that more efficient exploration will lead to a more efficient exploration step and a more effective $q_{\theta}$ for the same amount of compute.
>
> > Why did you not anneal in your MCMC?
>
> We employ MCMC to explore sequence transformations that improve the objective, and then use our learned $q_{\theta}$ to greedily maximize this objective. While annealing MCMC could help it attain a better local optima, this could actually be counterproductive to the goal of exploring productive sequence-to-sequence transformations. Compared to other works such as https://openreview.net/forum?id=ZMP0Bki9aK above, we seek to efficiently optimize using $q_\theta$ in as few steps as possible, while annealing would require a much larger number of steps.
>
> > “One could spend the compute training $q_{\theta}$ on running more MCMC.”
>
> While it is true that the time spent training $q_{\theta}$ could be used to run MCMC for longer, the training cost is a fixed cost that occurs offline, while running further iterations of MCMC is an online cost which would add more computational expense for each inference example. We emphasize that once $q_{\theta}$ is trained, it can be applied to new instances without further MCMC steps. Training $q_{\theta}$ allows for rapid inference on unseen examples, while MCMC requires an expensive sampling process for each inference example.

---

> > ### Comment · Reviewer_qZh3 · 2025-04-01
> >
> > This mostly addresses my concerns. I appreciate most discrete optimization methods assume a small alphabet; thanks! It would however still strengthen the paper to include the cost of training $q_\theta$ when running MCMC -- in the protein case you only want to find one optimum so online vs offline is not a relevant distinction. Maybe a good thing to test would be just: how many MCMC iterations would it take to reach the optima reached by your method reported in the paper, if it's even reached at all? 5x? 20x? But not crucial for this rebuttal.

---

> > > ### Author Response · Authors · 2025-04-07
> > >
> > > Thank you for the idea for this experiment. We agree that in cases where we optimize from the same starting point, this is an interesting question, and will plan to update our paper with this analysis.

---

### Official Review · Reviewer_6dee · 2025-03-12

**Overall Recommendation:** 4

**Summary:**

The authors propose a method to efficiently perform extrapolative generation, optimizing a property of interest.
Specifically, they train an autoregressive model to predict new sequences or states that enhance the desired property, using training samples obtained from MCMC. They evaluate their approach across three domains: sentiment optimization, protein engineering, and text anonymization.
To investigate whether the autoregressive model benefits from intermediate states, the authors conduct an ablation study on training episode design. They compare single-step training episodes with multi-step episodes, exploring different selection strategies — uniformly sampled states versus transitions that yield high relative improvement.

## Update after rebuttal
In the rebuttal, my questions have been adequately addressed. I still think this work as an interesting and valuable contribution to the conference, and I find this opinion to be somewhat supported by the other reviewers. Therefore, I will maintain my original score.

**Claims And Evidence:**

Yes, the claims made are well supported by convincing evidence.

Main claim/conclusion:
> We find that the autoregressive model can extrapolate as well or better than MCMC, but with the additional benefit of significantly higher sample efficiency.

The main conclusion is well supported by Table 1, Table 2, and Table 3. This is especially convincing due to the fact that the authors' approach has been evaluated on three different domains.

**Essential References Not Discussed:**

--

**Experimental Designs Or Analyses:**

* The experiments comparing the proposed approach to MCMC appear well-motivated and relevant.
* Autoregressive refinement: The results in this section seem a bit inconclusive. While intermediate state generation improves performance in the protein domain, its impact on anonymization and sentiment tasks remains unclear. Could the authors provide an intuition — or ideally, a deeper analysis — on when and why autoregressive refinement is beneficial in different scenarios? Additionally, when constructing training episodes, how should the hyperparameter controlling the number of included states be set or adjusted for different tasks?

**Methods And Evaluation Criteria:**

Experiments have been performed for the three different domains mentioned above. Experimental setups seem reasonable.

**Other Comments Or Suggestions:**

--

**Other Strengths And Weaknesses:**

### Other Strengths:
- Clarity: The paper is very-well written with a clear scope. Figure 1 and the toy example help ease the reader into the paper. The authors provide their rationale whenever needed to understand their assumptions and hypotheses.

### Other Weaknesses:
- Missing information on computational burden: While this may be a minor issue, it would have been helpful to include more details on the additional computational cost associated with model training and inference. If negligible, the reported differences in the needed iterations indeed should be a strong argument for the efficiency of the proposed method.

**Questions For Authors:**

See above.

**Relation To Broader Scientific Literature:**

The authors' work builds on [1]. Because of their relevant experimental setup and their good results, I think the authors provide a significant contribution to the community.

[1] Padmakumar, Vishakh, et al. "Extrapolative controlled sequence generation via iterative refinement." International Conference on Machine Learning. PMLR, 2023.

**Theoretical Claims:**

--

---

> ### Author Rebuttal · Authors · 2025-04-01
>
> We appreciate the detailed and thoughtful feedback, and aim to address remaining questions and concerns here.
>
> > “While intermediate state generation improves performance in the protein domain, its impact on anonymization and sentiment tasks remains unclear.”
>
> In the case of our sentiment task, there is no clear benefit to adding states: the task is sufficiently simple that a single edit is the most effective way to extrapolate, hence why we consider first/best to be the best method in this case. In more complex tasks we find it to be more important to have additional steps as there can be a generalization benefit, as in protein synthesis and anonymization. Additional edits consistently decrease semantic similarity for anonymization but improve EER up to a certain point, after which the SBERT decreases without consistent improvement in EER. We select our number of states based on this point.
>
>
> | Episode length | EER | SBERT|
> | ------------- | :-------------: |  :-------------: |
> |2 (first/best)| 0.132 | 0.923 |
> | 3 | 0.161 | 0.857 |
> | 4| 0.15 | 0.827|
> | 5 | 0.224 | 0.835 |
> | 6 | 0.187 | 0.776 |
> | 7 | 0.198 | 0.762 |
> | 8 | 0.2 | 0.745 |
>
>
>
>
> > Missing information on computational burden: While this may be a minor issue, it would have been helpful to include more details on the additional computational cost associated with model training and inference.
>
> Thanks for the suggestion! Indeed, the computational burden reported in our tables is purely inference-time computation. We recognize that our model has the additional computational burden of generating training data and fine-tuning a model on that data. However, this is a single fixed cost at training time. $q_{\theta}$ takes seconds for each inference example where MCMC takes ten minutes or longer for a single batch at inference time. The number of iterations in our tables are meant to provide an understanding of the scale of the ongoing computational cost at inference time, and does not account for the fixed (“offline”) training cost. We will be sure to include more details on the offline training costs in our revisions.

---

> > ### Comment · Reviewer_6dee · 2025-04-03
> >
> > Dear Reviewers,
> >
> > thank you for your answers. My questions have been addressed.
> >
> > I will maintain my original rating.

---

### Official Review · Reviewer_UTbQ · 2025-03-15

**Overall Recommendation:** 3

**Summary:**

This paper proposes an improvement to the MCMC algorithm, specifically the random search methods in the Monte Carlo exploration. Instead, a model trained from the MCMC searching trajectories is applied to greedily optimize the properties of interest. Empirically, the proposed method is able to sample efficiently and extrapolate well in natural language and biological tasks.

**Claims And Evidence:**

Most claims in the submission are supported by convincing evidence. However, there are several gaps in the proposed method and experiments to support the novelty and performance of this paper. For more comments, please refer to details in "Methods and Evaluation Criteria" and "Experimental Designs or Analyses" sections.

**Essential References Not Discussed:**

NA

**Experimental Designs Or Analyses:**

- It would be helpful to add an ablation of the model performance with different $q_{\theta}$ and compare to demonstrate the key components that lead to the best $q_{\theta}$.
- The paper does not compare to model based optimization methods in reinforcement learning, for example,, [1].

[1] Designing Cell-Type-Specific Promoter Sequences Using Conservative Model-Based Optimization. NeurIPS 2024.

**Methods And Evaluation Criteria:**

Most parts of the proposed methods and evaluation make some sense. However, there remain several issues:

- A theoretical perspective of why $q_{\theta}$ leads to more efficient sampling and what makes an optimal $q_{\theta}$ for the MCMC algorithm is missing.
- Although the proposed method effectively reduce the sampling iterations, the MCMC sampling to generate data for the training process of the model should also be considered, and an ablation on how the number of iterations for data generation in model training affect the model performance would be helpful.

**Other Comments Or Suggestions:**

NA

**Other Strengths And Weaknesses:**

The paper proposes an interesting method that utilizes a learned policy to replace stochastic exploration approaches. More theoretical perspectives on sampling efficiency and discussions on the method's extrapolation capability would strengthen the paper.

**Questions For Authors:**

NA

**Relation To Broader Scientific Literature:**

The paper is related to guided sampling and optimization methods, as well as applications in both natural language and biological domains.

**Theoretical Claims:**

There are no theoretical claims in this paper.

---

> ### Author Rebuttal · Authors · 2025-04-01
>
> We sincerely appreciate the feedback offered in this review, and hope to address some concerns.
>
> >”A theoretical perspective of…  what makes an optimal $q_{\theta}$  for the MCMC algorithm is missing.”
>
> We agree theoretical grounding is important. Nonetheless, compared to previous approaches, we believe our approach has demonstrated its robustness across three different settings, with minimal problem-specific tuning. Furthermore, though the learned extrapolation model $q_{\theta}$ lacks theoretical guarantees despite its empirical success, the MCMC search procedure that is the basis for fitting $q_{\theta}$ inherits the usual benefits of MCMC. Since extrapolative generation is an understudied area of deep learning, we hope our contribution can motivate more theoretical results in the future.
>
> > “...the MCMC sampling to generate data for the training process of the model should also be considered…”
>
> While we note the reviewer’s concern, we emphasize that although we use MCMC as a source of training data for $q_{\theta}$, MCMC is not required on new problem instances; $q_{\theta}$ can be applied to new starting sequences without MCMC if the extrapolation criteria are the same. Thus, besides the extrapolation benefits of $q_{\theta}$, it also effectively amortizes the sampling process.
>
> > “an ablation on how the number of iterations for data generation in model training….”
>
> We appreciate this suggestion and present our ablation of limited MCMC exploration on our anonymization task. We limit the MCMC chains to 50% and 25% of their length before constructing training episodes. Below, we show that increasing resources spent during data generation leads to improvements in our trained $q_{\theta}$ model. We consider this to be a strength of our approach; the theoretical properties of MCMC allow for more exploration and better fit of the target distribution as MCMC is run for more steps, offering a mechanism to improve generalization at the expense of further offline training time. In particular, the table below suggests that running MCMC for longer (e.g., 150%, 200%) could lead to better results than presented in the paper.
>
> | Exploration | EER | SBERT
> | ------------- | :-------------: | :-------------: |
> | 25% |0.2 | 0.701 |
> | 50%| 0.188 |  0.781|
> | 100% | 0.224| 0.835 |
>
> > “...compare to demonstrate the key components that lead to the best $q_{\theta}$…”
>
> Regarding ablations of $q_{\theta}$, thank you for the suggestion. We do analyze several important components of $q_{\theta}$ in the text of the paper, most notably the data selection strategy (Section 3.4) and reward choice (Appendix A). After following reviewer suggestions, we can now additionally present results for the effects of the number of iterations for data generation (see above) and for the effects of episode length on the anonymization task:
>
> | Episode length | EER | SBERT|
> | ------------- | :-------------: |  :-------------: |
> |2 (first/best)| 0.132 | 0.923 |
> | 3 | 0.161 | 0.857 |
> | 4| 0.15 | 0.827|
> | 5 | 0.224 | 0.835 |
> | 6 | 0.187 | 0.776 |
> | 7 | 0.198 | 0.762 |
> | 8 | 0.2 | 0.745 |
>
> We find that additional edits improve EER at the expense of decreasing semantic similarity; this is only true up to a certain point, after which the SBERT decreases without consistent improvement in EER, demonstrating the importance of this component in $q_{\theta}$. Multiple iterations are similarly useful for protein synthesis.
>
> We are happy to discuss additional factors should there be further concerns. For example, the paper as it stands focuses on a basic autoregressive architecture for $q_{\theta}$. However, it is likely that our results could be improved, for example by initializing from larger pre-trained models for $q_{\theta}$ or employing different training strategies.
>
> Regarding comparisons to further baselines, we agree this would strengthen the work. To our knowledge, our comparisons include SOTA extrapolative baselines for the challenging extrapolation tasks we consider, but of course there are many other tasks to consider, such as the design of cell-type-specific promoters for gene delivery considered in [1]. However, the complex approach described in [1] involves a very application-specific pipeline involving 5 different steps which would be hard to adapt to our more general setting. On the other hand, our approach is largely application-agnostic provided a suitable pre-trained LM is available.
>
> Regarding comparisons to RL-based methods such as [1], our approach may be viewed as performing off-policy reinforcement learning, even though we do not explicitly cast it that way (see e.g. https://arxiv.org/pdf/2106.02039). We considered using a more overt framing of the work as RL, but felt the additional notational baggage made it harder to understand the main idea. We view our contribution as a first effort to adapt off-policy RL to extrapolation settings by bridging RL and MCMC. More discussion of this point will be included in the paper.

---

> > ### Comment · Reviewer_UTbQ · 2025-04-07
> >
> > Thank the authors for the rebuttal. It resolves most of my concerns and I increased my score.

---

### Official Review · Reviewer_d3mh · 2025-03-15

**Overall Recommendation:** 4

**Summary:**

This paper presents a new approach for extrapolative sequence generation tasks, utilizing sequences produced through Markov Chain Monte Carlo (MCMC) exploration as training data. This approach targets tasks that necessitate the generation of new sequences exceeding previously recorded property values, such as in protein engineering, sentiment control, and text anonymization. The method initially utilizes MCMC to investigate and sample sequences that optimize specific target properties. The sampled Markov chains are employed to train an autoregressive model that predicts advantageous sequence transformations in fewer steps, thereby extrapolating beyond the original data distribution. This approach enhances sample efficiency by decreasing the number of necessary inference steps, while preserving sequence fluency and semantic coherence.

**Claims And Evidence:**

The evidence presented in the paper effectively supports the authors' claims. The experimental methodology is robust, and the results demonstrate significant advancements compared to previous methods in the specified context. The assertions regarding enhanced extrapolation capability and sample efficiency are supported by the data, with only minor caveats as discussed.

**Essential References Not Discussed:**

N/A

**Experimental Designs Or Analyses:**

The evaluation in the paper uses three benchmark tasks in distinct domains – protein engineering (ACE2 stability), text sentiment style transfer (Yelp reviews), and text anonymization. This selection is a strong point of the work: it covers both biological sequence generation and natural language generation with different kinds of target properties.

**Methods And Evaluation Criteria:**

The current benchmarks are essential and strong enough to support the claims.

**Other Comments Or Suggestions:**

1. The current framework typically utilizes initial sequences for transformation, such as the starting review or the original author’s text, that fall within the training distribution by design; for instance, a 3-star review may serve as a basis for generating a 5-star output. The system's performance under unusual initial states or noisy inputs has not been assessed. For example, in cases where a review is excessively lengthy or incorporates sarcasm, which may be underrepresented in the training data, can sentiment extrapolation maintain its reliability?

2. The diversity of the generated proteins from the current version remains unclear, which is significant in the context of protein engineering. In the context of model extrapolation, there exists a risk of mode collapse, wherein the model may identify and rely on a specific strategy or template to attain elevated performance across multiple outputs. The sentiment model may learn to append a standardized highly positive statement to the conclusion of each review to ensure a 5-star prediction, rather than genuinely altering the review in a diverse manner. The present assessment did not quantify the diversity or uniqueness of outputs.

**Other Strengths And Weaknesses:**

Weakness: a potential limitation of the approach itself is its dependence on the quality of the scoring function and MCMC samples. If the scorer is mis-specified or the MCMC exploration is very limited, the AR model will learn a suboptimal strategy. In principle, one could consider enhancements like adaptive sampling (to focus on more promising regions of state space) or use an ensemble/agreement of multiple scorers to mitigate bias.

**Questions For Authors:**

Please refer to the sections above.

**Relation To Broader Scientific Literature:**

N/A

**Theoretical Claims:**

This paper introduces the technical and theoretical part clealy.

---

> ### Author Rebuttal · Authors · 2025-04-01
>
> Thanks for the thoughtful response. We address some of the raised questions here.
>
> >”If the scorer is mis-specified or the MCMC exploration is very limited, the AR model will learn a suboptimal strategy…”
>
> These are valid concerns. A strength of our approach is that it benefits from the wealth of more sophisticated MCMC approaches; while we choose to use a basic sampler in this work, approaches such as the suggested adaptive sampling could lead to improved MCMC chains and thus better performance for $q_{\theta}$. We consider this to be a strength of our approach, as the theoretical properties of MCMC allow for more exploration and better fit of the target distribution as MCMC is run for more steps, offering a mechanism to improve generalization at the expense of further offline training time.
>
> > “... In the context of model extrapolation, there exists a risk of mode collapse…”
>
> We acknowledge that there is a possibility of mode collapse for extrapolation tasks, and that we should examine the diversity of our outputs. In consideration of this point, for our protein task, we counted the number of unique outputs, finding that 100% of the 10k proteins generated using $q_{\theta}$ were unique. For sentiment and anonymization, we ran corpus BLEU score between all pairs of generated sentences, finding that we achieve only 1.39 BLEU for sentiment and 0.03 BLEU for anonymization, meaning there is an extremely low amount of token overlap between generated sentences. Furthermore, in Appendix E we show randomly selected examples of our generated outputs for sentiment and anonymization, which demonstrate significant sample variance. In light of this evidence, we believe it is unlikely that mode collapse is occurring.

---

### Decision · Program_Chairs · 2025-05-01

**Decision:**

Accept (poster)

**Comment:**

All reviewers agree with the acceptance. It has a solid contribtion.